A nonlinear total variation based computed tomography (CT) image reconstruction method using gradient reinforcement

Ertas Metin ertas@istanbul.edu.tr
Department of Electrical and Electronics Engineering, Istanbul University-Cerrahpasa , Istanbul , Turkey
Góes-Neto Aristóteles
Electronic publication date: 2024 Jan 8
Publication date: 2024
Volume: 12
Electronic Location ID: e16715
Received 2023 May 30; Accepted 2023 Dec 4
Copyright: ©2024 Ertas
Copyright year: 2024
Copyright holder: Ertas
License: This is an open access article distributed under the terms of the Creative Commons Attribution License, which permits unrestricted use, distribution, reproduction and adaptation in any medium and for any purpose provided that it is properly attributed. For attribution, the original author(s), title, publication source (PeerJ) and either DOI or URL of the article must be cited.
License URL: https://creativecommons.org/licenses/by/4.0/

Keywords: Total variation, Gradient reinforcement, Iterative image reconstruction, Few-view imaging

Funding: The authors received no funding for this work.

==============================
Compressed sensing-based reconstruction algorithms have been proven to be more successful than analytical or iterative methods for sparse computed tomography (CT) imaging by narrowing down the solution set thanks to its ability to seek a sparser solution. Total variation (TV), one of the most popular sparsifiers, exploits spatial continuity of features by restricting variation between two neighboring pixels in each direction as using partial derivatives. When the number of projections is much fewer than the one in conventional CT, which results in much less sampling rate than the minimum required one, TV may not provide satisfactory results. In this study, a new regularizer is proposed which seeks for a sparser solution by reinforcing the gradient of TV and empowering the spatial continuity of features. The experiments are done by using both analitical phantom and real human CT images and the results are compared with conventional, four-directional, and directional TV algorithms by using contrast-to-noise ratio (CNR), signal-to-noise ratio (SNR) and Structural Similarity Index (SSIM) metrics. Both quantitative and visual evaluations show that the proposed method is promising for sparse CT image reconstruction by reducing the background noise while preserving the features and edges.

Introduction

Dose reduction while keeping the image quality at its maximum has been one of the leading problems in computed tomography (CT) image reconstruction. With the availability of compressed sensing theory, the exact reconstruction of images became possible with a sampling rate far less than the Nyquist rate (Donoho, 2006). This phenomenon opened a new era called sparse reconstruction in medical imaging. Sparse reconstruction in tomographic imaging can be achieved by acquiring fewer projections, using a limited view angle, or a combination of these two. To solve these ill-posed inverse problems, iterative reconstruction has been widely used. The iterative reconstruction method can be interpreted as a balance between the data and model fidelity terms (Gunturk & Li, 2013). Data fidelity term forces dependence on the projections through a defined forward operator while the model fidelity works as regularization to enforce the prior information to narrow down the solution set of the iterative reconstruction problem. One of the simplest but strongest ideas is using actual image denoiser algorithms as a prior in model fidelity. In the literature, block matching 3D (BM3D) (Dabov et al., 2007), total variation (TV) (Sidky, Kao & Pan, 2006; Velikina, Leng & Chen, 2007), non-local denoiser operators, Buades, Coll & Morel (2005) and their modified versions adapted to the specific imaging problems, Sghaier et al. (2022); Jin et al. (2010); Kim et al. (2016); Zhang et al. (2021) have been integrated as model fidelity for iterative reconstruction.

Total variation has been one of the most used regularization terms in iterative sparse image reconstruction, especially in tomographic (Sidky, Kao & Pan, 2006; Velikina, Leng & Chen, 2007; Chen et al., 2013) and tomosynthesis imaging (Fränkel et al., 2013; Mota et al., 2020). TV has been used in image reconstruction problems due to its convex nature, its strong ability to preserve edges and recover image discontinuities which is the main concern in sparse tomographic reconstruction as the comprehensive sampling of the entire Fourier space is not available. It has been shown in the literature that the sparse image reconstruction performance is dramatically increased not only by classical isotropic TV but also changing the representation of classical TV in various ways, such as: by adding new weights or new gradient terms and combining these two methods (Hu & Jacob, 2012; Candès & Boyd, 2008; Xie et al., 2020; Pang et al., 2019).

Adding different weights to each partial gradient used in the cost function enables the isotropic form of TV to be used as an anisotropic form (Chen et al., 2013; Jin et al., 2010). The partial gradients were also weighted depending on their directions concerning the feature directions in the images and resulted in better orientation preservation in textural imaging (Bayram & Kamasak, 2012).

The impact of single direction constraint over a limited angle tomographic image taken from both x and y directions also showed that the directional information plays an important role in feature preservation in the image taken from a limited view angle (Zhang et al., 2021). There have been alternative modified gradient based total variation algorithms (Zhang et al., 2021; Sakurai et al., 2011; Fan Liao & Shu, 2015; Huang et al., 2016; Huang et al., 2018; Hu & Jacob, 2012; Zhou & Fan, 2021). Sakurai et al. proposed a faster four directional TV algorithm that brings two additional partial terms to the horizontal and vertical directional terms in classical TV (Sakurai et al., 2011). Another directional total variation algorithm was proposed to eliminate the directional artefacts while preserving the edges, features and fine details in the limited view angle tomographic reconstruction (Huang et al., 2016). They used the idea that the artefacts in the limited scan angle are aligned with the direction of this limitation in view angle. By enforcing the partial gradient in the scan direction where the directional artefacts mostly occur, they achieved a superior result than the conventional TV (Huang et al., 2016).

Fewer sampling in sparse CT reconstruction weakens neighborhood relations in the spatial domain. Therefore the imaging problem in sparse CT becomes severely under-determined. Though traditional TV has been successfully used in CT imaging along with iterative methods such as simultaneous algebraic reconstruction technique (SART), it is still desired to be improved for sparse CT imaging. Therefore a better regularizer which is sparser than the traditional TV is desired. Motivated from these facts and the success of these re-derived variants of the TV algorithm, a new regularizer is proposed specifically for sparse-view tomographic imaging which will be called as reinforced TV (rTV) from now on. Though the traditional TV also exploits the continuity of features in spatial domain to some extent, the proposed model handles it more efficiently thanks to its reinforced gradients feature in both axial and sagittal directions. This reinforcement is done by adding an additional gradient term to each partial gradient in the same direction as the traditional TV. The traditional TV uses one neighboring pixel to be used in the gradient calculation for each partial derivative; however, in the proposed method, another neighboring pixel adjacent to the gradient direction is considered in the cost function and the gradient is still calculated on the central pixel where the derivation is taken on.

Next section will give detailed information about classical TV and the proposed reinforced TV with its derivation and implementation. The following section represents the experiment setup and the results acquired from both Shepp-Logan and human CT data. The results will be compared by the Structural Similarity Index (SSIM), signal-to-noise ratio (SNR) and contrast-to-noise ratio (CNR) of specific regions with overall visual assessments. The discussion section will try to answer some open questions which might arise with the implementation and application of the proposed method. Finally, the paper is concluded with a short brief and future expectations.

Materials and Methods

The forward model used in tomographic imaging is represented as follows: (1) y=Ax+η

where, x ∈ RN is the image to be reconstructed in a vector form, A ∈ RMxN is the forward operator which represents the X-ray tracing coefficients calculated by Siddon’s algorithm (Siddon, 1985), η ∈ RM is the reconstruction noise and y ∈ RM is the observation data; in other terms the projection data in the tomographic reconstruction. In sparse-view imaging the observation number M is not enough to fully reconstruct the image x; in other words M <  < N. Because of this deficiency in projection data, sparse-view image reconstruction becomes a severe ill-posed imaging problem. Considering the sparse-view tomographic imaging, total variation regularized iterative reconstruction was proven to be an efficient way to solve severely underdetermined inverse problems (Sidky, Kao & Pan, 2006). The following optimization problem is used in tomographic reconstruction: (2) x ˆ=argminx∥y−Ax∥2+λ∥Rx∥

where, the first term is the data fidelity of the objective function which controls the distance from the observation space. The second term is the norm of the regularization term, ∥R(x) ∥ which narrows down the solution set adhering to the data fidelity by using the prior knowledge about the image x. λ is the regularization parameter balancing these two powerful forces. x ˆ is the optimal final image. Considering the nature of medical images, minimizing the total variation of the image results in a smoother background with sharper edges and more visible details. The total variation of an image x can be represented as follows: (3) XTV= ∑iI ∑jJxi,j−xi+1,j2+xi,j−xi,j+12

with, (4) Dh=xi,j−xi+1,jandDv=xi,j−xi,j+1.

Dh and Dv are the discrete partial gradients along x and y directions respectively. I and J represents the number of pixels in horizontal and vertical directions respectively. Since medical images are made up of large organs, bone structures, and tissues; by reducing the gradient magnitude the piece-wise smoothness will be well-preserved. Though one pixel of neighborhood relation has been proven to be effective in sparse image reconstruction, additional neighboring pixels in gradient calculation would increase the denoising performance. Thus in this study, a reinforced total variation (rTV) algorithm is proposed, in which the effect of the central pixel from which the derivative is taken on is increased by enhancing the neighborhood degree in each partial gradient operator. The reinforced gradient nature of the proposed algorithm also enhances the locality level in the objective function and increases the denoising performance. The gradient estimate in classical TV is replaced by D ˜h and D ˜v with one additional neighboring contribution and rTV is formulated as follows: (5) XrTV= ∑iI ∑jJD ˜h2+D ˜v2

where, (6) D ˜h=xi,j−xi+1,j+xi,j−xi+2,j=2xi,j−xi+1,j−xi+2,jD ˜v=xi,j−xi,j+1+xi,j−xi,j+2=2xi,j−xi,j+1−xi,j+2.

The difference between Eqs. (3) and (5) is visually represented in Fig. 1. The additional term in the partial gradients is represented as a blue outer link between the neighboring pixels. In order to solve the minimization of Eq. (5), the gradient descent method is used due to its simplicity. Another representation of the location of the pixels in an image that contributes to the gradient information is shown in Fig. 2. The blue-colored pixels constitute the gradients in x and y directions used in Eq. (5). The most important step in the gradient descent method is the derivative approximation of the cost function in Eq. (5). Here, the derivative approximation of Eq. (5) is shown numerically in Eq. (7) and schematically in Fig. 2 so that both the derivation of the proposed cost function and similar TV cost functions can be easily implemented by the readers.

(7) ∂xrTV∂xi,j=22xi,j−xi+1,j−xi+2,j+22xi,j−xi,j+1−xi,j+22xi,j−xi+1,j−xi+2,j2+2xi,j−xi,j+1−xi,j+22+ɛ−2xi−1,j−xi,j−xi+1,j2xi−1,j−xi,j−xi+1,j2+2xi−1,j−xi−1,j+1−xi−1,j+22+ɛ−2xi−2,j−xi−1,j−xi,j2xi−2,j−xi−1,j−xi,j2+2xi−2,j−xi−2,j+1−xi−2,j+22+ɛ−2xi,j−2−xi,j−1−xi,j2xi,j−2−xi,j−1−xi,j2+2xi,j−2−xi+1,j−2−xi+2,j−22+ɛ−2xi,j−1−xi,j−xi,j+12xi,j−1−xi,j−xi,j+12+2xi,j−1−xi+1,j−1−xi+2,j−12+ɛ.

Figure 1 The schematic representation of gradient information of TV and rTV.

Left: TV Right: rTV (see the original manuscript for colorful representation).

Figure 2 The coordinate relation of xi,j with its neighboring pixels in a 2D image for TV derivations.

(see the original manuscript for a colorful representation).

In Eq. (7), ɛ is a very small number to eliminate singularities in the equation. Two specific conditions are required for a function to be minimized by gradient descent: convexity and differentiability. The main idea is to iteratively calculate the next point by using the gradient at the current position over the function until it reaches the global minimum. The main consideration in the gradient descent method is to set the step size. In this study, an adaptive step size increasing with each step k in the order of 2k, is used. By this adaptive selected step, the optimal point can be reached much faster, and in the case of jumping around the optimal point, the step size is reduced by 1 and the same process is followed until it reaches the closest point with a minimum error value or a specific iteration number.

Steps in Eq. (7) are also visually represented on an image in Fig. 2 for easier reproducibility of the results. In Fig. 2, five pixels are numbered from 1 to 5 in brown color. When the pixels in the image in Fig. 2 are derived from xi,j, only these five pixels will contribute to the derivative expression. Since all other pixels are independent of the xi,j pixel considering the cost term in Eq. (5), it will be appended to the expression as 0 to Eq. (7). Thus, pixels represented from 1 to 5 in brown color in Fig. 2 will contribute to the derivative approximation in Eq. (7) as corresponding order from the top row to the bottom row, respectively.

In this study, regularizers are integrated into the algebraic reconstruction technique (ART). The steps of the ART+rTV algorithm are described in Algorithm 1 as a flow for the sake of easier implementation and reproducibility of the results. The stopping criterion for the proposed method can either be controlled by iteration number or a certain error value between two consecutive reconstructed images.

_______________________ Algorithm 1 ART+rTV______________________________________________________________________ Require: x0      =     0,niter     =     20,ntviter     =     20,λ    =     1,α    =    1,i=1,k=0,learningrate = 10−6,ɛ = 10−4,a = 1   while k < niter do       procedure ART(argmin     x    ∥y − Ax∥22)                ⊳ Image update by ART            if i ≤ nproj then               xk+1 = xk + αyi−〈Ai,xk〉   ∥Ai∥2     ATi                           ⊳ ART formulation                 i + +            end if       end procedure       xart ← xk+1          procedure RTV(argmin     x    ∥x − xart∥22)                   ⊳ rTV Regularization            n = 0            x0rTV  ← xart              while a > ɛ and  n ≤ ntviter do               Grad = −2(y − AxnrTV ) + λ∂xn rTV_  ∂xi,j                    m = 0                 while the cost decreases do                   xn+1 rTV  ← xn rTV − (2m)learning rateGrad                   m + +                 end while               a ←∥xart − xn+1 rTV ∥         n + +            end while       end procedure       k + +   end while___________________________________________________________________________

Resolution in the simulations is chosen as 512 × 512. All simulations are performed on NVIDIA GeForce RTX 2080 Ti GPU card using MATLAB 2020b. Half scan angle (π) is used in image acquisition of 30, 60 and 90 equally spaced and uniformly distributed projections with 6°, 3°, and 2° intervals respectively. λ parameter is chosen as 1 to keep a balanced relation between data and model fidelity terms to avoid competition between these two terms. Since iterative methods generally converge to the desired image level to a large extent before the 10th iteration with negligible change in metric values, the number of iterations in this study was limited to 20 for both ART and TV in the experiments.

Results

Experimental setup ad performance metrics

In this section, simulated Shepp-Logan phantom and human CT data are used to validate the proposed rTV algorithm. The proposed method rTV was compared with a widely used iterative reconstruction method ART and TV for Shepp-Logan phantom and ART, TV, directional TV, and 4D-TV for human CT data.

In sparse tomographic reconstruction, artefacts commonly arise from the nature of the problem due to discontinuities in the sparse selection of projections as it uses only few projections, unlike the full-dose CT, where the sampling rate meets the Nyquist rate and the artefacts come mostly from quantum noise of X-ray photons (Poisson) and electronic (Gaussian) noise of detector. Thus, the projections are not artificially noised by Gaussian and Poisson noise for Shepp-Logan phantom experiments in this study.

Since the human CT data projections are already noised during the observation, no additional noise is considered in the present experimental setup. The numerical comparisons are done by using several imaging metrics such as SNR, SSIM and CNR. The formulation of the signal-to-noise-ratio is given as follows: (8) SNRdB=10log10PsignalPnoise.

Another metric to be used in the comparisons is the SSIM (Structure SIMilarity) index which mimics the human vision and scales this resemblance to a numerical value between 0 and 1 (Wang et al., 2004). The higher the score is, the higher the similarity between the noise-free image and the reconstructed image is. SSIM metric basically measures the perceptual difference between two images by using the combination of contrast, luminance and structural similarities between images and is formulated as follows: (9) SSIMa,b=2μaμb+C1+2σab+C2μa2+μb2+C1σa2+σb2+C2

where a and b are images whose similarities will be measured, μa and μb are the average pixel values of images a and b respectively, σab is the cross covariance of images a and b, σa2 and σb2 are the variances and finally C1 and C2 are the variables which balance the equation. The final metric used in the comparison is the contrast-to-noise-ratio (CNR) whose formulation is given as follows: (10) CNR=μfeature−μbackgroundσbackground

where, μfeature and μbackground are the mean pixel intensity values of the feature and background respectively. σbackground is the standard deviation of the background which shows how smooth the background is.

Experimental results

In order to validate the performance of rTV, a number of experiments are performed on both Shepp-Logan phantom and human CT data. Figure 3 shows the reconstructed images on Shepp-Logan phantom of FBP, ART, TV and rTV with 30, 60 and 90 projections. FBP algorithm with 30 projections shows the worst results with severe artifacts and distortions all over the phantom compared with 60 and 90 projections as expected. Though image quality in FBP improves as the number of projections increases, the performance of FBP is still not enough to recover fine details with smoother background. ART is able to improve the performance of FBP to some extent though it is not yet quite satisfactory. Though, TV suppresses the artefact arising from ART, it still suffers from background noise since the lack of projections results in discontinuities in the projection domain. However, as the projection number increases the fine details become more visible and distinguishable in TV. The best visual results are achieved via rTV method as the exact reconstruction of the original Shepp-Logan phantom is almost achieved by rTV with 90 projections with a negligible difference. Moreover, rTV with 60 projections result is also very close to the reconstructed image of TV with 90 projections as the background noise is similar with visible fine details. Thus, the proposed method seemed to be more powerful in reducing background noise while keeping the fine, small details and edges in better shape.

Figure 3 Shepp-Logan reconstruction results.

(A) Original phantom, (B to D) FBP results with 30, 60 and 90 projections, (E to G) ART results with 30, 60 and 90 projections, (H to J) TV results with 30, 60 and 90 projections, (K to M) rTV results with 30, 60 and 90 projections.

SNR, SSIM and CNR values for TV and rTV are given in Table 1. Numerical results for FBP and ART are not given in Table 1 since the performance suffers from severe artefacts. The results show that SSIM values of the proposed method gives better results in all projection numbers. However, SNR values at 90 projections show that TV gives slightly a higher SNR value. In order to take a deeper analysis, the change of SNR values for each projection numbers are given in Fig. 4. It is known that the noise level increases as the number of projections decreases. As the proposed algorithm rTV is a powerful tool that considers an additional gradient term to the minimization problem, the increased noise level favors the proposed method. As a result, the SNR graphs show an increasing difference between 30 and 60 projection numbers as the iteration number increases with a wider difference in 30 projections since the noise level is higher in 30 projections. But the SNR graphs of reconstruction with 90 projections are almost overlapped each other. At this point, the visual results show that the rTV shows a better denoised image with higher SSIM but slightly lower SNR than TV.

Table 1 Shepp-Logan phantom SNR, SSIM and CNR results of TV and rTV with 30, 60 and 90 projections.

	TV 30	TV 60	TV 90	rTV 30	rTV 60	rTV 90	
SNR	20.36	27.07	32.73	20.9	27.43	32.31	
SSIM	0.960	0.984	0.996	0.982	0.995	0.998	
CNR	4.55	7.27	14.26	5.75	10.08	17.49	

Figure 4 SNR graphs of Shepp-Logan phantom results per iteration.

Top left: 30 projections, top right: 60 projections, bottom left: 90 projections.

Since SNR value is significantly depended on mean squared error (MSE), CNR could also play an important role in the analysis. For CNR analysis, a small region of the reconstructed Shepp-Logan phantom is zoomed in to take a closer look for visual evaluation and CNR calculation. The zoomed-in regions are shown in Fig. 5. Zoomed-in images give a better understanding of the reconstruction performance of the proposed method. Both methods suffer from a significant noise in the reconstructions with 30 projections. The best feature recovery is achieved via rTV with sharper boundaries and a much smoother background which makes the features more detectable. SNR values of TV reconstruction with 90 projections were slightly higher than rTV, however, the zoomed-in images show that the features are almost similar to the original image with less background noise in rTV reconstruction with 90 projections. The numerical values of CNR are also correlated with the visual evaluations as the CNR value of rTV methods is much higher than the one of TV. The highest CNR value is achieved via 90 projections as expected with a significant difference. Thus, the CNR values support that the proposed method recovers images not only with less background noise but also with sharper contrast differences between features at boundaries.

Figure 5 Zoomed in region of reconstructed Shepp-Logan phantom.

(A) Original phantom and region of interest (ROI) in yellow, (B to D) TV results with 30, 60 and 90 projections, (E to G) rTV results with 30, 60 and 90 projections.

Shepp-Logan has been a global image for performance analysis of tomographic reconstruction however, the performance of the proposed method should also be evaluated on human CT data for its clinical potential. A human CT image is used for the performance analysis of the proposed rTV method. A similar outline used in Shepp-Logan analysis is also followed in human CT analysis. In addition to the conventional TV, the proposed rTV method is also compared with 4D-TV, (Sakurai et al., 2011) and directional TV, (Huang et al., 2016) on human CT data. Though both methods have also been implemented with faster optimizers with specific optimization considerations in the literature; in order to achieve a fair comparison, the minimization of these two methods is performed with the same gradient descent method used in the rTV algorithm. The directional TV in Eq. (11) was aimed to be used in limited view angle reconstruction only, thus it is not directly an alternative to the proposed method. The related formulation used in directional TV is shown in Eq. (11) and 4D-TV in Eq. (12). (11) XdirectionalTV= ∑iI ∑jJxi,j−xi+1,j2+xi,j−1+xi,j−xi,j+1−xi,j+22

(12) X4DTV= ∑iI ∑jJxi,j−xi+1,j2+xi,j−xi,j+12+xi+1,j−xi,j+12+xi,j−xi+1,j+12.

The original image and the reconstructed images are given in Fig. 6. FBP provides the least successful reconstruction performance as the noise level is relatively high compared to the other methods in all projection numbers. The reconstruction performance also positively changes with the increased projection number as expected. Reconstruction with 30 projections recovers images with severe background noise as the phantom is more complex in human CT than Shepp-Logan. Moreover, the background noise over breast regions in images reconstructed with 60 projections is still dominant and makes the image look blurry though other regions are relatively more noise-free. To our knowledge, the only acceptable image for performance analysis is the one reconstructed with 90 projections as the number of projections below 60 comes up short of a reliable analysis due to a high level of undersampling. The best reconstruction performance is achieved with sharper edges and smoother background with less noise via the proposed rTV method with 90 projections. Moreover, the vessels in the heart such as the pulmonary artery and the ascending part of the aorta are well preserved with smoother and homogenous background with sharper edges in the proposed rTV method than the other methods. Since the scan angle is not a concern in this sparse view reconstruction and the artifacts are not direction-dependent in the human CT image used in the reconstruction, the directional TV did not perform at its best compared to the one in the limited view angle CT reconstruction. However, it still performs better than the conventional TV. 4D-TV also shows very close results with the proposed method in terms of edge preservation. However, with its powerful denoising feature, breast tissue with less noise in all 5 methods is observed in the images reconstructed with the proposed rTV method.

SNR and SSIM values of human CT data for FBP, ART, TV, directional TV, 4D-TV, and rTV are given in Table 2. The full dose CT image taken from the Deep Lesion dataset is used as a ground truth image for the analysis. The numerical results are also well-aligned with the visual evaluations as rTV with all projection numbers results in better SNR and SSIM values than all other methods. The highest values of SNR and SSIM were achieved at images reconstructed with 90 projections. Since the phantom itself is more complex than Shepp-Logan, the performance of the proposed method for real data is much more obvious as the difference between each SNR and SSIM value for each specific projection number is more than what it is in the Shepp-Logan phantom. This difference can be easily seen when both Tables 1 and 2 are examined together. The change of this gap per iteration number between SNR values of ART, TV, directional TV, 4D-TV, and the proposed rTV can also be seen in Fig. 7. The gap between SNR values per iteration number increases and gets wider as the iteration number increases and more importantly shows an increasing trend instead of flatness. When we remove ART in Fig. 7, it is clearly observed that SNR value of the rTV algorithm will differ more from the other methods, and the gap becomes wider as the number of iterations increases. This proves that the proposed method shows even better performance for images with higher complexity and real noise.

Figure 6 Human chest CT reconstruction results.

(A) Original CT image, (B to D) FBP results with 30, 60 and 90 projections, (E to G) ART results with 30, 60 and 90 projections, (H to J) TV results with 30, 60 and 90 projections, (K to M) directional TV results with 30, 60 and 90 projections, (N to P) 4D TV results with 30, 60 and 90 projections, (R to T) rTV results with 30, 60 and 90 projections.

Discussion

One of the main challenges of the proposed method is its increased computational complexity. The CPU-based reconstruction time of 512×512 Shepp-Logan phantom for 20 iterations is 1.31, 1.83, 1.84, 1.85 s for TV, directional TV, 4D TV and rTV respectively. Though adding an additional term to the partial gradients increases the reconstruction performance, it also increases the minimization time as the derivation of the proposed cost function becomes more complex. However, rTV also convergences much faster than the conventional TV, thus the same performance might be achieved in fewer iterations. To overcome this time cost, alternative faster optimizers such as ADMM can be used (Boyd et al., 2011). As the regularization algorithm is suitable for parallel processing, the reconstruction time can further be reduced by using a GPU-based processor.

Another limitation of the proposed method is the minimum resolution required to be able to be applied for clinical applications. As the resolution decreases, the number of pixels representing tiny features in the image will be fewer than the one required for the assumption that the proposed method relies on. In order to see the optimum resolution, the proposed method is compared with the conventional TV for 128, 256, 512 and 1024 pixel resolutions for 60 projections and Table 3 shows the SNR change as the resolution increases. As the resolution gets smaller than 512×512 pixels more and more, the proposed method starts performing poorer than the classical TV due to the invalidation of the theory that the proposed idea relies on. Thus the proposed method is preferable for resolution values not much smaller than 512×512 since the uniformity between adjacent pixels will decrease as the resolution gets smaller. Commercial CT scanners mostly use 512×512 pixel resolutions. In addition to this, the proposed method’s superiority becomes even more visible as the resolution increases thanks to its reinforced gradient feature as expected.

Table 2 SNR and SSIM results of FBP, ART, TV, rTV, directional TV and 4D-TV with 30, 60 and 90 projections for CT image.

	SNR	SSIM	
	FBP	ART	TV	Direc.TV	4D TV	rTV	FBP	ART	TV	Direc.TV	4D TV	rTV	
30 proj.	4.60	15.19	16.18	16.47	16.52	16.72	0.308	0.850	0.873	0.880	0.880	0.888	
60 proj.	10.28	20.15	22.71	23.20	23.45	23.99	0.462	0.930	0.950	0.956	0.956	0.965	
90 proj.	14.08	23.65	27.88	28.12	28.93	29.18	0.623	0.962	0.979	0.981	0.983	0.988	

Figure 7 SNR graphs of human chest CT results per iteration.

Top left: 30 projections, top right: 60 projections, bottom left: 90 projections.

Table 3 SNR results of TV and rTV for 128, 256, 512 and 1024 resolutions.

		128×128	256×256	512×512	1024×1024	
SNR	TV	43.05	31.19	27.04	23.61	
rTV	38.1	30.25	27.43	24.96	

Though there have been studies addressing the appropriate selection of λ, the impact of regularization depends on many features that change during the reconstruction. As a result, the appropriate selection of λ is still an open question. In this study, the lambda is set to a fixed value of 1 to create a balanced relation between the fidelity and regularization terms. However, setting an adaptive lambda parameter instead of a fixed one would yield better results with an even faster convergence rate and better detail preservation. But for a fair performance comparison, these assumptions are left unanswered.

The clinical usability of the proposed reconstruction algorithm is another topic to be discussed. The performance of the proposed method is shown to produce better results than the traditional iterative method ART, TV, 4D-TV, and directional TV by using visual assessment, SNR, and SSIM which makes the proposed method a good alternative in clinical use. However, radiologists’ assessments should also be required to validate the numerical results. A group of radiologists could also assess the reconstructed images in the sense of their clinical usability and score them in accordance with their performance and this might come up with a clinical score that can also be used as an alternative metric. This can be further integrated into future clinical-based studies to see whether the algorithm is favorable to be used in the clinic.

The rTV formulation in Eq. (5) can be easily adapted to digital breast tomosynthesis (DBT) or 3D volumetric imaging methods by adding an additional gradient term in the z-direction. However, the resolution in the z-axis should be taken into account when updating Eq. (5). For example; In DBT imaging, the resolution in the z-axis is in the order of mm compared to the µm resolution in the xy-axes. This approximately 10-fold difference in the different axes must be adapted to the model by adjusting the weights of the gradient terms in the three axes to match the resolutions. On the other hand, for a volumetric imaging method with nearly or equal resolution in all directions, a similar gradient reinforcement in the z direction can be applied directly.

Conclusion

In this study, a new reinforced total variation algorithm was proposed which reinforces the gradient along both axial and sagittal directions for sparse view iterative tomographic reconstruction. The proposed method shows a significant noise removal performance method by increasing the locality and neighborhood relation. The performance analysis of the proposed method, rTV is performed on analytical phantom and human chest CT images by comparing with classical ART, TV, directional TV, and 4D-TV both qualitatively by visual assessments, and quantitatively by SNR, SSIM and CNR calculations.

Supplemental Information

Supplemental Information 1 Source Code

Click here for additional data file.

Supplemental Information 2 Raw data for Fig. 4

Click here for additional data file.

Supplemental Information 3 Raw data for Fig. 4

Click here for additional data file.

Supplemental Information 4 Raw data for Fig. 4

Click here for additional data file.

Supplemental Information 5 Raw data for Fig. 7

Click here for additional data file.

Supplemental Information 6 Raw data for Fig. 7

Click here for additional data file.

Supplemental Information 7 Raw data for Fig. 7

Click here for additional data file.

Additional Information and Declarations

Competing Interests

Author Contributions

Data Availability

The authors declare there are no competing interests.

Metin Ertas conceived and designed the experiments, performed the experiments, analyzed the data, prepared figures and/or tables, authored or reviewed drafts of the article, and approved the final draft.

The following information was supplied regarding data availability:

The DeepLesion data is available at the NIH-owned Box repository: https://nihcc.app.box.com/v/DeepLesion (images.png).

The code is available in the Supplemental File and at GitHub: https://github.com/metinertass/rTV.

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
