# Peer review of "A nonlinear total variation based computed tomography (CT) image reconstruction method using gradient reinforcement"

_PeerJ, doi:10.7717/peerj.16715_

## Round 0.1 · original submission · Major Revisions

We have received two reviews about your article, and both are positive. Although one of the reviewers indicated only minor revisions, the other one indicated major revisions. Therefore, it is very important to address the points of the latter, especially those related to hyperparameter tuning (ε parameter, regularization parameter, and number of iterations). Please, go through all these points, answer them and modify your paper accordingly.

Reviewer 1 ·

Basic reporting

The author propose an alternative approach to the well known Total Variation (TV) regularisation for Sparse CT tomography reconstruction. The reinforced TV (rTV) formulation considers one additional grid point in both the horizontal and vertical directions. This extension increases the local information acquired during the iterative process and experimentally has more benefits both visually and in terms of quality metrics.

Experimental design

The author provide three experiments on both simulated and experimental datasets. It would be nice to perform a simple benchmark on dataset designed for limited angle tomography, see the Helsinki Tomography Challenge https://www.fips.fi/HTC2022.php. Although, it is quite common to use a Shepp-Logan phantom for these comparisons, in this particular case the reconstructions as well as the quality metrics do not reflect a realistic scenario.

Validity of the findings

It is clear from both the report of quality metrics and visual comparison that rTV outperforms other classical TV definitions. However, there are some alternatives in the context of combining differently finite difference discretizations, that can easily be tested, presented and referred. For instance, one could explore the anisotropic definition of TV weighted by the same regularization parameter or different along the directions. Also, eq15 in "Approximating the Total Variation with Finite Differences or Finite Elements" or the weighted difference between anisotropic and isotropic definition of TV, "A Weighted Difference of Anisotropic and Isotropic Total Variation Model for Image Processing."

Additional comments

L19: analitical phantom
L26: far less than the Nyquist rate , Donoho (2006).
L35: block matching 3D (BM3D), Dabov et al. (2007). Check for commas when you cite a paper.
L60: proposed to get rid of the directional artefacts. “get rid of” is an informal expression
L66-L69: The sentence is not clear
Figure 3 and 5: The captions in Figure 3 and 5 are very confusing. It is better to label cols/rows or add some text inside the image.
L 155-158: could you elaborate more in these sentences?
Table 1: The SSIM values in this table are unrealistic because they indicate clearly that the ground truth image is recovered.
L 193: An analytic reconstruction (FBP/FDK) or SIRT/ART with low number of iterations is important as a reference for comparison.
Algorithm 1: Do you use a non-negativity constraint after the ART step? How do you enforce positive reconstruction.
L 293: “T”he rTV formulation.

Also, there is no discussion about the hyperparameter tuning e.g., ε parameter, regularization parameter and number of iterations.

Reviewer 2 ·

Basic reporting

1. Figures:

Please include a color bar with the images for clearer comprehension by readers. Where relevant, also add a scale bar.

For figure 4, enhancing the font size of axis labels, titles, and legends would improve readability.

Experimental design

1. System Details:

Imaging: To provide a complete picture, it would be valuable if the author could share either a schematic or photograph of the imaging system. Can details of the X-ray imaging system be provided?

For equation (1), it would be beneficial if the author can provide the X-ray coefficient details in the forward operator A

Reconstruction: Which optimizer has been employed across all the reconstructions?

Validity of the findings

1. Baseline for Reconstruction Methods:

Every reconstruction method requires a baseline for evaluation. It would be beneficial if the author could include FBP results or other appropriate results to serve as this baseline.

2. Evaluation of Computational Cost:

Could the author compare the reconstruction times across TV, rTV, Direct TV, and 4D-TV? Assessing computational costs is critical, especially when real-time reconstruction is vital. With methods like the current deep learning approach and FBP offering quicker reconstruction times than iteration-based methods, it would be advantageous for the author to consistently include FBP results in all comparative evaluations, both in terms of image quality and reconstruction time.

3. Inclusion of Other Regularizers:

I recommend incorporating other regularizers, such as the l1 norm, for a clearer understanding of the strengths and weaknesses of the TV regularizer.

4. Effect of compression ratio:

The paper currently evaluates results using 30, 60, and 90 projections (indicative of different compression ratios). How would the outcomes change with an increased number of projections, perhaps 180? A comparison of image quality across different compression ratios would be insightful.

---

## Round 0.2 · accepted · Accept

Both reviewers are fully satisfied with the current form of the manuscript and informed that all the issues were solved; therefore, my final editorial revision is that this manuscript is accepted for publication.

Reviewer 1 ·

Basic reporting

I would like to extend my appreciation to the author for addressing all of my comments and suggestions provided during the review. In terms of the direction TV and limited angle, you can have a look at https://arxiv.org/abs/2310.01671. In the future, it would be nice to see how your proposed method performs in the limited angle case.

Experimental design

All comments and suggestions are resolved.

Validity of the findings

All comments and suggestions are resolved.

Additional comments

All comments and suggestions are resolved.

Reviewer 2 ·

Basic reporting

no comment

Experimental design

no comment

Validity of the findings

no comment